# Comparison of TEC Calculations Based on Trimble, Javad, Leica, and Septentrio GNSS Receiver Data

**Vladislav Demyanov [1,2,\*], Maria Sergeeva [3,4], Mark Fedorov [2], Tatiana Ishina [2], Victor Jose Gatica-Acevedo [3,5] and Enrique Cabral-Cano [6]**

[1] Department of near-Earth Space Physics, Institute of Solar Terrestrial Physics, Siberian Branch of Russian Academy of Sciences, 664033 Irkutsk, Russia

[2] Department of transportation control systems, Irkutsk State Transport University, 664074 Irkutsk, Russia; metalgear@mail.ru (M.F.); t.v.ishina@yandex.ru (T.I.)

[3] SCiESMEX, LANCE, Instituto de Geofisica, Unidad Michoacan, Universidad Nacional Autonoma de Mexico, Morelia 58089, Mexico; maria.a.sergeeva@gmail.com (M.S.); vic_gatica@hotmail.com (V.J.G.-A.)

[4] CONACYT, Instituto de Geofisica, Unidad Michoacan, Universidad Nacional Autonoma de Mexico, Morelia 58089, Mexico

[5] Instituto Politecnico Nacional, Mexico City 07738, Mexico

[6] Instituto de Geofisica, Universidad Nacional Autonoma de Mexico, Mexico City 04510, Mexico; ecabral@geofisica.unam.mx

\* Correspondence: demyanov_vv@irgups.ru; Tel.: +7-950-051-3095

**Abstract:** A Global Navigation Satellite System (GNSS) receiver is, to some extent, a "black box" when its data is used for ionospheric studies. Our results based on Javad, Septentrio, Trimble, and Leica GNSS receivers have proven that the accuracy of the slant Total Electron Content (TEC) calculation can differ significantly depending on the GNSS receiver type/model, because TEC measurements depend on the carrier phase tracking technique applied in a receiver. The correlation coefficient between carrier phase noise in L1 and L2 channels is considered as a possible indicator that shows if the L1-aided tracking technique or independent tracking is applied inside a receiver. An empirical model of the TEC noise component was provided to determine the TEC noise value in different types/models of GNSS receivers.

**Keywords:** GNSS receiver; slant TEC; Trimble; Javad; Leica; Septentrio

## 1. Introduction

Slant Total Electron Content (TEC) calculations are based on data from carrier phase and code delays of GNSS satellite signals received by dual-frequency ground-based GNSS receivers. TEC and its derivatives are widely used for different tasks of ionospheric studies. The most popular TEC-based indices are ROTI, DROTI, AATR, DIX, and DIXSG [1–4]. The accuracy of these indices (and eventually their interpretation) depends on the quality of the primary TEC measurements.

Today, temporal resolution of GNSS receiver output data can reach up to 100 Hz [5]. Such a high temporal resolution allows us to detect small-scale weak ionospheric turbulences and provides a TEC measurement accuracy of approximately $10^{-3}$ TECU or even better [6,7]. High-rate GNSS data can help researchers answer a fundamental question regarding which sampling rate is the border between the weak ionospheric events and non-informative noises [8]. At the same time, in regard to the problem of detecting weak ionospheric disturbances, signal processing techniques inside a GNSS receiver play the same crucial role as the data temporal resolution. Unfortunately, a GNSS receiver is, to some extent, a "black box" for researchers. Recent studies have proven that the TEC and TEC-based indices can differ significantly when derived from different types/models of GNSS receivers and based

on different ionosphere-free linear combinations. Yang and Liu [9] observed a difference in TEC and ROTI values calculated from data of L2P(Y) and L2C GPS observables. McCaffrey et al. [10] showed that independently tracked carrier phase dynamics was significantly more accurate than the L1-aided observables. Padma and Kai [11] defined the optimal ionosphere-free combination for dual-frequency receiver's L1, L2C, and L5 GPS signals in terms of sensitivity and observation noise.

In our opinion, it is important to define the pure noise component of TEC in relation to the signal processing technique and the type of ionosphere-free linear combination, as well as receiver hardware. The aim of this study was to reveal how the noise component of the slant TEC depends on the receiver type/model. Our research tasks included the following: (1) to unfold if the ionosphere-free L1-L2 linear combination is the L1-aided technique product in each receiver type; (2) to determine the overall phase noise level in each receiver considering it as a "black box". The experiment and modeling results were used to solve these tasks. Carrier phase data of GPS L1 and L2 frequencies from Javad, Trimble, Leica, and Septentrio receivers were used for the analysis.

## 2. Materials and Methods

The high-rate carrier phase data of GPS L1 and L2 frequencies were obtained from five GNSS receivers of different types/models. The data files are available as Supplementary Materials at (S1), (S2), (S3) and (S4) links below. Table 1 provides the details of the experiment. The dates of observations were chosen based on the availability of high-rate data for our research. Geomagnetic conditions were defined by the daily-averaged Kp-index and the daily minimum Dst-index values.

**Table 1.** GNSS receivers whose data was involved in the analysis and details of the experiment.

| Receiver Type/Model | Station | Time Resolution | Type of Observable | Site | Coordinates | | | | Date, Year | Kp, Dst |
| --- | --- | --- | --- | --- | --- | --- | --- | --- | --- | --- |
| | | | | | Geographic | | Geomagnetic | | | |
| | | | | | Lat,° | Lon,° | Lat,° | Lon,° | | |
| Javad DELTA | ISTP | 50 Hz | L1CA, L2P | Irkutsk, Russia | 52 | 104 | 42.36 | 177.21 | 19 April 2018 | 1, -6 nT |
| Trimble NetR9 | UCOE | 20 Hz | L1CA, L2C, L2P | Mexart, Coeneo, Mexico | 19.8 | −101.68 | 27.88 | 31.11 | 19 April 2018 | 1, −6 nT |
| Trimble NetR9 | SPIG | 20 Hz | L1CA, L2C, L2P | San Pedro Martir, Mexico | 31.03 | −115.45 | 37.56 | 47.35 | 10 April 2018 | 1, −6 nT |
| Septentrio POLARX5S | LEUV | 50 Hz | L1CA, L2P | Leuven, Belgium | 50.84 | 4.73 | 51.83 | 89.33 | 19 April 2018 | 1, −6 nT |
| Leica GR10 | IPN1 | 50 Hz | L1CA, L2P | Mexico City, Mexico | 19.29 | −99.64 | 27.54 | 28.89 | 18 February 2020 | 2, −52nT |

The ionosphere-free linear combination of two different frequencies allowed us to calculate the slant TEC along the line of sight between the satellite and receiver as follows [6]:

$$I_S = \frac{1}{40.308} \frac{f_1^2 \cdot f_2^2}{f_1^2 - f_2^2} [L1 \cdot \lambda_1 - L2 \cdot \lambda_2 + const + \sigma\phi + n\phi] \tag{1}$$

where $\lambda_1$, $\lambda_2$, $L1$, and $L2$ are the wavelengths and carrier phase counts (including integer and fractional parts of the phase cycles) at $f_1$ and $f_2$ GNSS frequencies; *const* is the unknown constant due to the phase ambiguity; $\sigma\varphi$ is the sum of TEC calculation errors; and $n\varphi$ is the sum of the unpredicted phase noise components from the L1 and L2 linear combination which is usually ignored.

The instantaneous phase range value ($\Phi_i = L_i \cdot \lambda$) for the *i*-th time point is defined as follows [12]:

$$\Phi_i = \rho_i - I_i + T_i + \delta m_i + c\big(dt_i(t) - dt(t - \tau_i)\big) + c\big(\delta_i(t) + \delta(t - \tau_i)\big) + \lambda N_i + \varepsilon_i \tag{2}$$

where $\rho_i$ is the geometric range "Satellite Vehicle (SV)-receiver"; $I_i$, $T_i$, and $\delta m_i$ are the ionospheric, tropospheric, and multipath errors; $c\cdot(\ldots.)$ are the components that describe SV and receiver clock offsets; $\lambda N_i$ is the phase ambiguity and $\varepsilon_i$ is the unpredicted carrier phase noise.

All the components of Equation (2), except $\varepsilon_i$, define the SV motion trend, satellite and receiver clock offsets, as well as slow random refractive variations of the carrier phase due to regular ionosphere, troposphere, and quasi-regular signal to multipath fading. All these errors form the sum of the total error term, $\sigma\varphi$, Equation (1). Similarly, the final component, $\varepsilon_i$, can be represented as the sum of unmodulated carrier phase noises such as ionospheric amplitude and phase scintillations, tropospheric phase rapid variations, SV oscillator anomalies, GNSS receiver thermal noises, Allan deviation, and vibration-induced noises.

The components of $\varepsilon_i$ which have a variation period less than 0.1 s (>10 Hz) are usually considered to be the non-informative phase noises [13]. It is known that the instantaneous carrier phase is obtained from the phase-lock loop (PLL) filter based on the discrete Markov's chain model at each $i$-th time point as follows [12]:

$$\phi_i = \phi_{i-1} + T_{COR} \cdot \frac{d\phi_{i-1}}{dt} ; \frac{d\phi_i}{dt} = \frac{d\phi_{i-1}}{dt} + T_{COR} \cdot \frac{d^2\phi_{i-1}}{dt^2} ; \frac{d^2\phi_i}{dt^2} = \frac{d^2\phi_{i-1}}{dt^2} + \varepsilon_i \tag{3}$$

where $T_{COR}$ is the PLL predetection integration time and $\varepsilon_i$ is the zero-mean Gaussian noise (i.e., the carrier phase noise term from Equation (2).

Considering Equation (3), the noise component $\varepsilon_i$ can be calculated from the second order derivative of the carrier phase, if the time resolution is higher than 10 Hz [13]. Consequently, standard deviation of the TEC noise component (i.e., $n\varphi$ from Equation (1) can be estimated as a sum of standard deviations of the phase noise components $\varepsilon_i$, at L1 and L2 frequencies. In this case, the covariance between these noises should be taken into account [14]:

$$\sigma_{n\phi} = \sigma_{L1} + \sigma_{L2} + 2R_{L1,L2} \cdot \sigma_{L1} \cdot \sigma_{L2} \tag{4}$$

where $\sigma_{L1}, \sigma_{L2}$ are standard deviations of the noise components at L1 and L2 frequencies; $R_{L1,L2}$ is the correlation coefficient between the phase noise at these two frequencies.

Some types of navigation receivers use the L1-aided technique [15] to track the phase of the signal at second frequency. There is an obvious advantage of this technique from the radio engineering point of view. Indeed, in the case of signal tracking failure in L2 or L5 channels, it is possible to realize the fastest signal relock and tracking in these channels. In contrast, this technique is not satisfactory if a GNSS receiver is used for ionospheric studies. Radio propagation effects depend on signal frequency, but the artificial connection between the phase and frequency measurements at L1 and L2, or L5 frequencies, results in the fact that the ionospheric effects of radio propagation are not observed correctly [10].

Usually, a researcher does not know which L2 signal tracking technique is implemented in a particular model/type of navigation receiver. We suppose that the correlation coefficient between the phase noise components in L1 and L2 channels can help to solve this uncertainty. The high correlation between phase noises in L1 and L2 channels argues for the L1-aided technique implementation in a particular receiver model/type. In contrast, the moderate or low correlation argues for the independent phase and frequency measurements in L1 and L2 channels, which is obviously better for the ionospheric studies.

To evaluate the TEC noise component and the impact of interchannel noise correlation on the final TEC value, an analytical model of the TEC noise component $n\varphi$ is suggested as follows:

$$n\phi_{MODEL} = \frac{\sigma_{n\phi}}{k} \cdot \frac{cf}{2\pi \cdot 10^{16}} \cdot \exp(-\gamma \cdot a) + n_0 \tag{5}$$

where $\sigma_{n\phi}$ is defined by Equation (4) from the experimental data processing; $n_0$, $k$, and $a$ are the model parameters; $c$ is the speed of light in vacuum; $f$ is the carrier frequency; and $\gamma$ is the SV elevation angle.

Equation (5) presents the statistical empiric model without physical interpretation of the model form and its parameters. It means regression analysis of the experimental data on the plane "TEC noise, elevation". The model was developed in two steps as follows:

1. Definition of the analytical form that adequately describes the trend of the experimental data on the plane "TEC noise, elevation". Usually, such a dependence is an exponential form [16]. Hence, we used the exponential form as an analytical base of the model.

2. Definition of the model parameters is based on the Legendre principal. We defined the model (Equation (5)) parameters by means of least squares method based on the experimental data on the plane "TEC noise, elevation". It was done by means of the equation linearization and involving the standard Newton method [17].

The TEC noise component is extracted from the slant TEC measurements by detrending it with the sliding window filter with 10 Hz cut-off frequency. It was revealed in [18] that 10 Hz temporal resolution was the approximate border between the pure noise components (thermal noises, Allan deviation, multipath noise, ionospheric scintillations, and receiver vibration), and lower frequency processes (regular and slow ionospheric and tropospheric refraction, reference oscillator long-term instability, Doppler frequency shift and drift). Noise component differences, considering the interchannel noise correlation, is determined as follows:

$$\Delta n\phi = n\phi 1_{MODEL} - n\phi 0_{MODEL} \tag{6}$$

where $n\varphi 1_{MODEL}$ and $n\varphi 0_{MODEL}$ are the TEC noise model (Equation (5)) assuming $R_{L1,L2} \neq 0$ (defined from the experiment) and $R_{L1,L2} = 0$ respectively.

## 3. Results and Discussion

Figures 1–4 show the diurnal variation picture for each receiver from Table 1. Each dot corresponds to the hourly-averaged value of a parameter obtained from data of the particular SV. The elevation angle values are also hourly averaged.

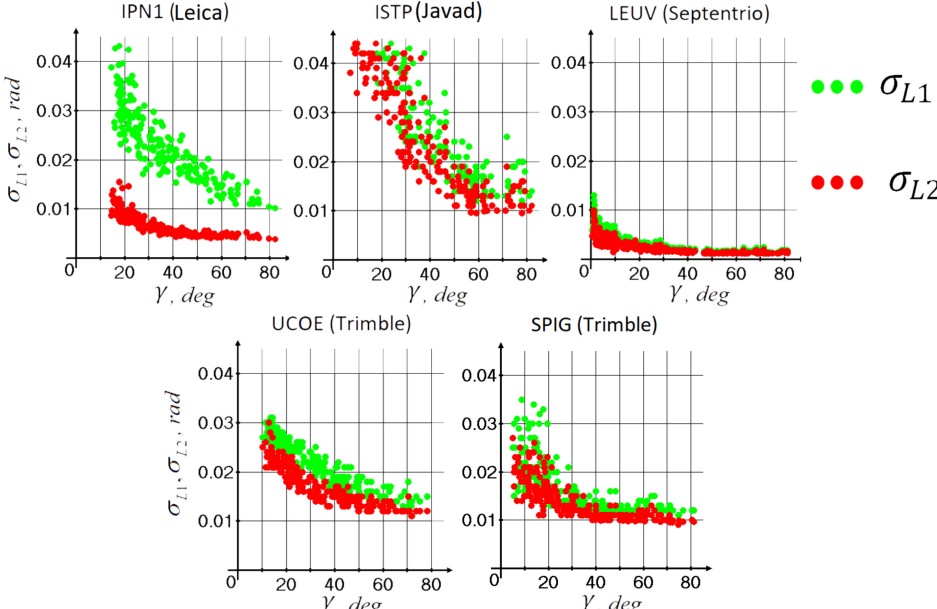

**Figure 1.** Dependence of standard deviation of the L1 and L2 carrier phase noise on the satellite elevation angle.

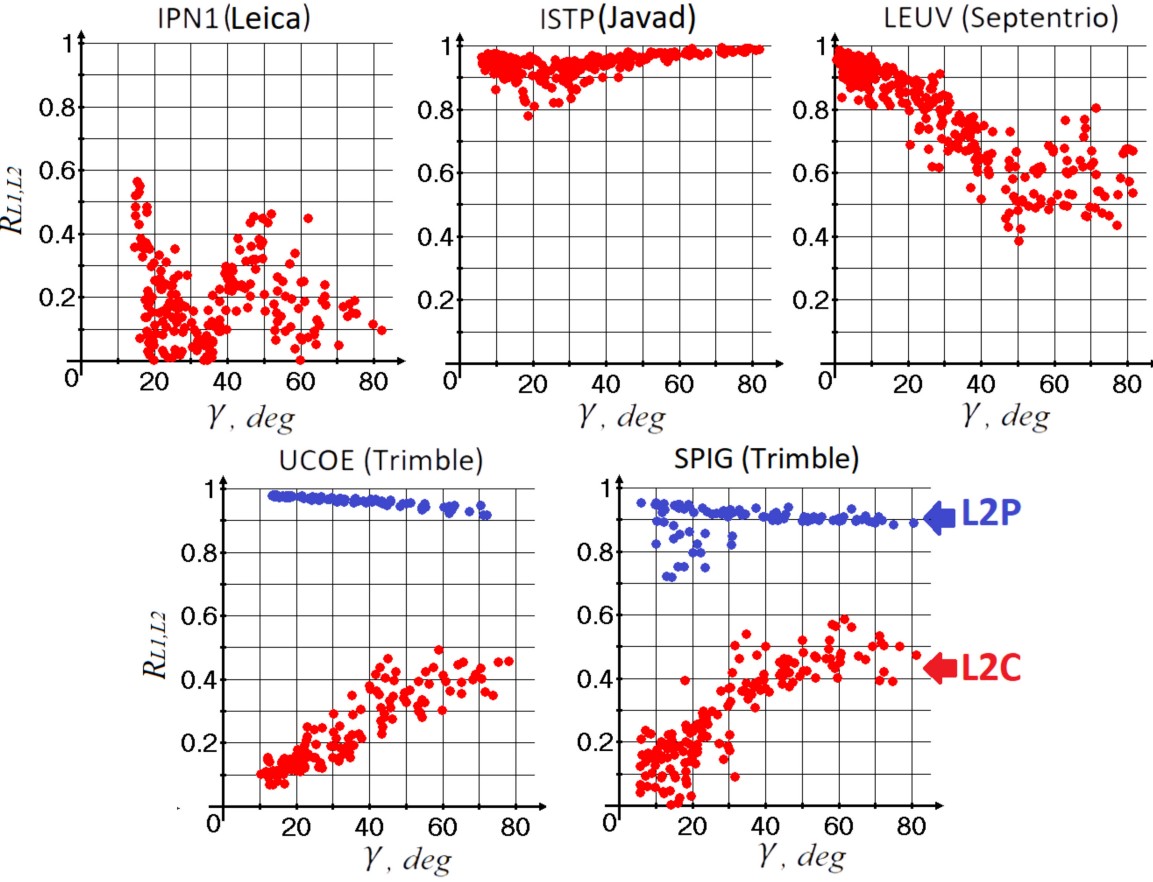

**Figure 2.** The correlation coefficient between the L1 and L2 carrier phase noises with respect to satellite vehicle (SV) elevation angle and receiver type.

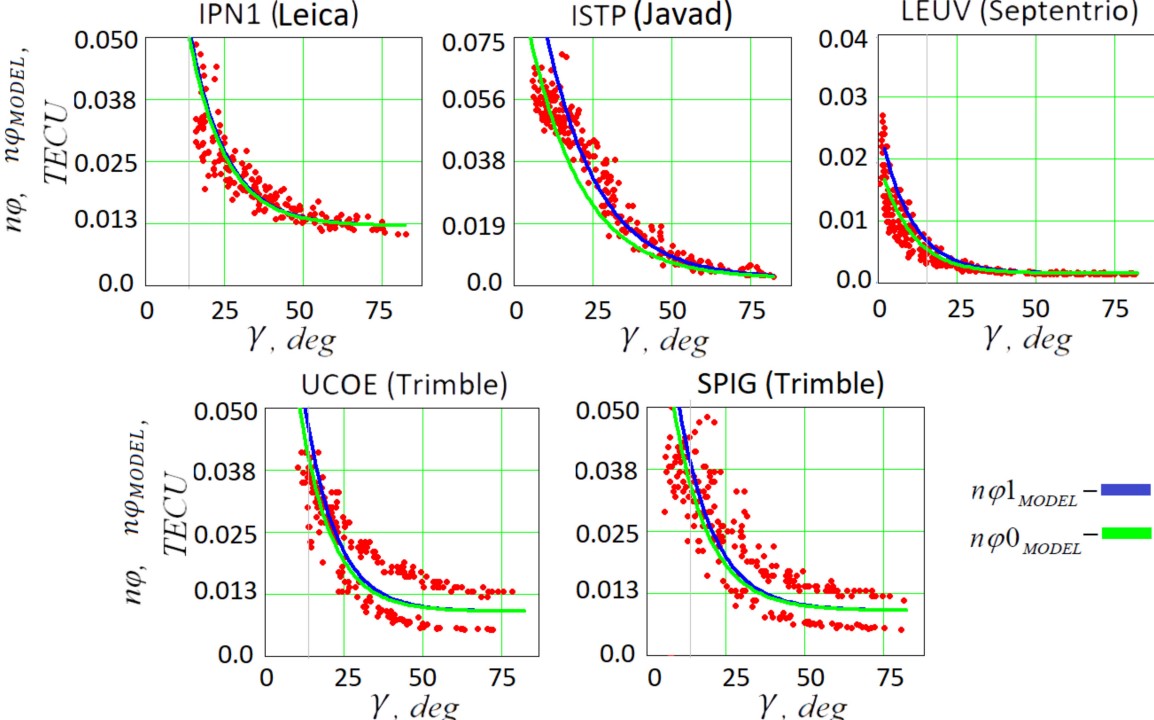

**Figure 3.** Total electron content (TEC) noise component dependence on the elevation angle ($n\varphi$, red dots and $n\varphi_{MODEL}$, blue and green curves).

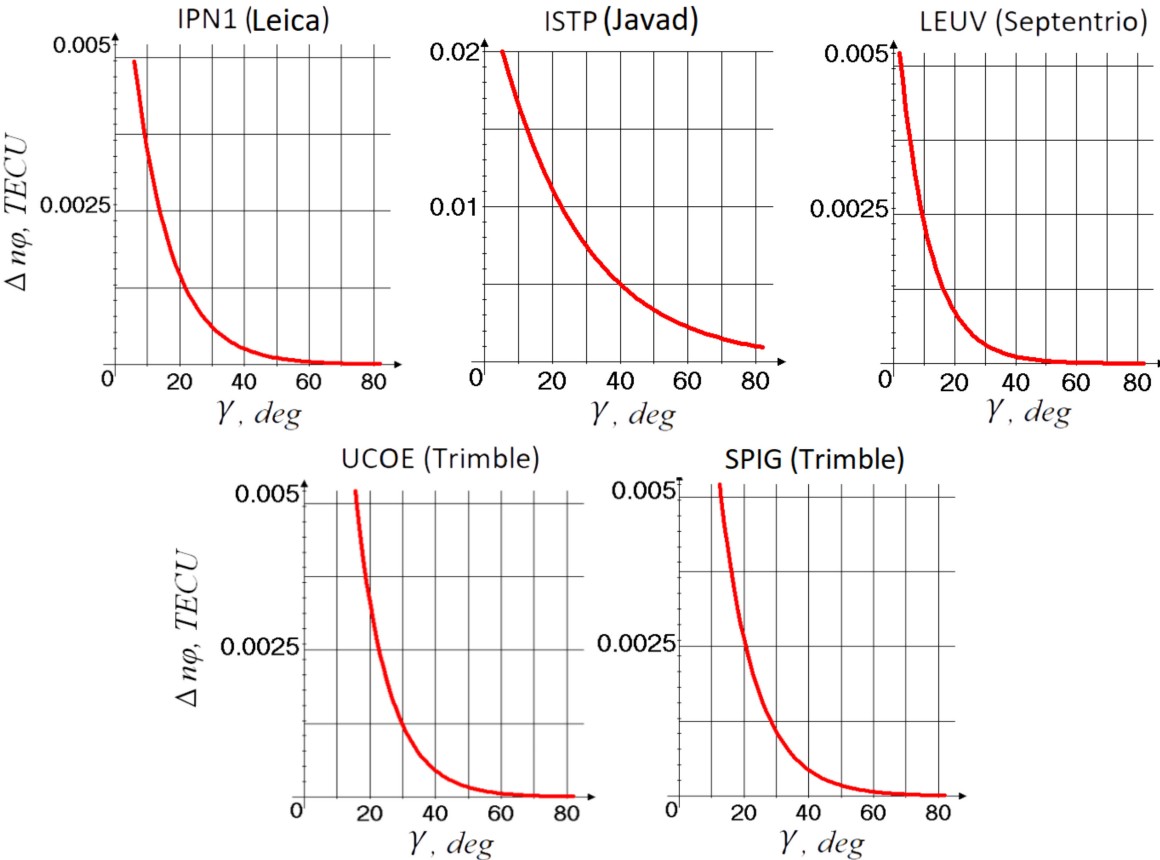

**Figure 4.** The difference between TEC noise components taking into account the interchannel noise correlation and receiver type.

Figure 1 shows the standard deviation of L1 and L2 carrier phase noises dependence on the satellite elevation angle. Each panel shows the results obtained for a particular type of GNSS receiver from Table 1. The L2C data were available only from the Trimble receivers. Thus, the upper panels show the results based on L1CA and L2P data and the lower panels show the results based on L1CA, L2P, and L2C data.

It is seen that the $\sigma_{L1}, \sigma_{L2}$ values and their overall dependence tendency significantly differ for receivers of different types, but the presence of this dependence on the SV elevation is clear in all cases. Usually, this typical dependence is explained by multipath and tropospheric impacts, as well as by the signal power fading at the low elevation angles. In addition, we note that the forms of the dependence, shown in Figure 1, significantly differ from one receiver type to another.

The highest phase noise level was observed for the Javad receiver at both L1 and L2 frequencies. In contrast, the Septentrio receiver showed the lowest noise level at both frequencies. In addition, for the Septentrio case, the difference between noises at L1 and L2 is negligible. The Leica receiver demonstrated the most "exotic" feature. There is a two-times difference in the carrier phase noises at L1 and L2 frequencies. There is nothing of the kind relating to other receiver types in this study. Such a difference between background phase noise in L1 and L2 channels cannot be explained by the weak magnetic disturbance that occurred on the day of the experiment (Table 1). Indeed, such a difference should have $\sim 1/f^2$ dependence and this proportion cannot be deteriorated with the ionospheric turbulences and scintillations, unless a frequency coherence disruption happens between L1 and L2 signals.

Figure 2 shows the correlation coefficient between the L1 and L2 carrier phase noises with respect to the SV elevation angle and receiver type. The highest correlation coefficient ($R_{L1L2} > 0.8$) was between the phase noise in L1 and L2 channels in the Javad receiver. It probably means that the

L1-aiding technique is used to track L2P signal inside this receiver type. The correlation coefficient for the Septentrio receiver is mostly higher than 0.5. This fact implies that probably the L2P tracking process is based on the L1-aiding technique inside the Septentrio receiver as well. Indeed, according to information by the Septentrio team, the cross-correlation is used for L2P signal tracking, but it is not used for L2C signal tracking (this data was not available in our case). In contrast, the correlation coefficient is mostly lower than 0.5 for the Leica receiver. It probably proves that the L1-aiding technique is not applied to track L2P signal inside this receiver type.

The most illustrative result was obtained from both Trimble receivers that were able to track both L2P and L2C signals, separately (Figure 2, two lower panels). The correlation coefficients between the noise terms of L2P and L2C carrier phase components are very different. There is no doubt that the L1-aiding technique is used to track L2P signal (blue dots) and it is not used for L2C signal tracking (red dots).

Another interesting and unexpected result is the correlation coefficient dependence on the satellite elevation. Different receivers demonstrated very different forms of this dependence. This issue is the subject for future research.

Figure 3 illustrates the noise component of TEC at L1 and L2 frequencies obtained from both the TEC measurements and by means of modeling. The blue curve represents the modeling results assuming that $R_{L1,L2} \neq 0$ ($n\varphi 1_{MODEL}$). The green curve represents the modeling results assuming zero correlation between these channels ($n\varphi 0_{MODEL}$). The red dots present the TEC noise obtained from the experiment. The upper panels (IPN1, ISTP, and LEUV) show the results based on the L1CA and L2P data and the lower panels (Trimble) show the results based on the L1CA, L2P, and L2C data.

The results, in Figure 3, are similar by their form and values to the characteristics shown in Figure 1 for all the receiver types and demonstrate direct relations between the phase noise at L1 and L2 frequencies and TEC noise. In general, the smallest TEC noise value for the Septentrio receiver and the highest value for the Javad receiver are observed. It is especially interesting that different signal tracking procedures for the L2P and L2C components result in different TEC noise components from the Trimble receiver (Figure 3, two lower panels). This result is in accordance with the results in Figure 2. The plots for the Leica and Trimble receivers are rather similar in form. However, the lowest TEC noise value of ≈0.013 TECU was observed for the Leica receiver. It is a rather high noise level, although the L1-aiding technique is not used to track the L2P signal in the Leica receiver (Figure 2, upper left panel). Probably this is due to the high phase noise level in the L1 channel of the Leica receiver (Figure 1, upper left panel).

Additionally, Figure 3 proves the overall good quality of the model (Equation (5)) describing the real TEC noise dependence on the elevation angle for all the receiver types. In the case of Trimble data (two lower panels), there is uncertainty in the model parameter definition. There is a difference between the TEC noise components derived from the L2Y and L2C signals due to the difference in the L2Y and L2C signal tracking procedures. Therefore, the model parameters (Equation (5)) were defined for an averaged trend of both of these noise components.

According to Equation (4), the presence of correlation can increase the standard deviation of the combined noise, $n\varphi$, depending on the correlation coefficient $R_{L1,L2}$. Figure 4 illustrates that the interchannel correlation of the L1 and L2 phase noise components yields a negligible impact on the TEC noise component in the case of elevation angles >20°. The largest value, $\Delta n\varphi \approx 0.01$ TECU, was observed for the Javad receiver at 20° elevation angle. This value is close to the sensitivity threshold of the TEC-based methods of the ionospheric disturbance detection [7]. The lowest value, $\Delta n\varphi \approx 0.00075$ TECU, was observed for the Septentrio receiver at the same elevation angle.

We should note that the real twice difference in TEC noise values derived from L2P and L2C signals (Trimble, Figure 3, lower panels) do not match the results in Figure 4 (two bottom panels). It probably means that the TEC noise is affected by the correlation between L1 and L2 carrier phase noises and also by other mechanisms depending on the receiver architecture. For example, the thermal noise of a receiver can also have an impact on the final TEC noise figure. The main sources of

thermal noise of a receiver are antenna and preamplifier. These hardware components are different for different types of receivers. In addition, the thermal noise depends on the environmental conditions at a certain location [18]. The carrier phase noise also depends on the carrier-to-noise ratio at the phase lock loop (PLL) input in each receiver. In addition, such hardware components as cable, filter, and analog-to-digital convertor play a certain role in the total noise picture.

## 4. Conclusions

Knowledge about the signal processing technique and the structure of a particular GNSS receiver is important for the correct interpretation of fine TEC fluctuations and, especially, for the ionospheric scintillation analysis. The signal processing technique inside the receiver can be revealed using the data of the carrier phase noises at L1 and L2 frequencies. In this study, the second-order derivative of the GNSS signal phase was used to extract the carrier phase noise from L1 and L2 data avoiding additional complex processing.

The results based on Javad, Septentrio, Trimble, and Leica GNSS data proved that the noise level of the slant TEC value can differ significantly if using different types/models of GNSS receivers for TEC reconstruction. The correlation coefficient between carrier phase noise in L1 and L2 channels indicates whether the L1-aided tracking technique or independent tracking is applied inside a receiver. According to our results, the L1-aided tracking technique is used in Javad, Septentrio, and Trimble receivers to track L2P signals. The Trimble receiver processes the L2C component without the L1-aiding technique implementation. The feature of the Leica receiver is the high carrier phase noise level in the L1 channel which is a limitation of the lower border of the TEC noise from this receiver type output. The overall comparison of four receiver types in regard to the ionospheric studies showed the best results for the Septentrio receiver type/model. The smallest TEC noise and carrier phase noise values at L1 and L2 frequencies were both observed for this receiver type. We use data from two Trimble and one Leica receivers installed in Mexico. According to our earlier works [19,20], we suppose that conditions over the close to each other UCOE (Trimble) and IPN1 (Leica) receivers are very similar during quiet geomagnetic periods. The conditions over SPIG (Trimble), northwestward from the two mentioned receivers, are rather different. At the same time, our study shows that the UCOE and SPIG results (Trimble) were similar but the IPN1 results (Leica) differed significantly. This proves again that receiver type plays an important role in TEC noise forming.

Finally, we should note that TEC noise is affected by the correlation between L1 and L2 carrier phase noises and also other mechanisms depending on the receiver architecture. This important issue is the subject for the future research.

**Supplementary Materials:** The datasets analyzed in this study are available in the following repositories: (S1) Demyanov Vladislav, Sergeeva Maria, Gatica-Acevedo Victor Jose, and Cabral-Cano Enrique (2020), GPS 50 Hz dataset for the case study entitled "Comparison of TEC calculations based on Trimble, Javad, Leica, and Septentrio GNSS receiver data", Part 1 (Version RINEX 2.11 and ASCII text files) (dataset), Zenodo, http://doi.org/10.5281/zenodo.3996442; (S2) Maria A. Sergeeva (2020), GPS 50 Hz dataset for the case study, Part 2 (Version sbf) (dataset), Zenodo, http://doi.org/10.5281/zenodo.3999204; (S3) Cabral-Cano Enrique, Salazar-Tlaczani Luis, Pérez-Enríquez Román, Sergeeva Maria (2020), very high rate GPS observables at Coeneo, Mexico (UCOE) continuously operating station, dataset for the case study Part 3 (dataset), Zenodo, https://doi.org/10.5281/zenodo.4002090; (S1) Cabral-Cano Enrique, Salazar-Tlaczani Luis, Servicio Sismológico Nacional working group, Sergeeva Maria (2020), very high rate GPS observables at San Pedro Martir, Mexico (SPIG) continuously operating station, dataset for the case study, Part 4 (dataset), Zenodo, https://doi.org/10.5281/zenodo.4002090. Kp-index values were obtained from ftp://ftp.swpc.noaa.gov/pub/indices/old_indices and Dst-index values were obtained from http://wdc.kugi.kyoto-u.ac.jp/dstdir/index.html.

**Author Contributions:** V.D. developed the conceptualization of this work; V.D. and M.S. designed the experiments; M.F. and T.I. developed the model code and performed the simulations; V.J.G.-A. and E.C.-C. performed the experiments; all the authors participated in the data processing and the analysis of the results; V.D. and M.S. prepared the manuscript with contributions from all the authors. All authors have read and agreed to the published version of the manuscript.

**Funding:** The main part of this work (conceptualization and methodology, theoretical foundation, modeling, and experimental results interpretation) was performed under the Russian Science Foundation grant no. 17-77-20005. The processing of the high-rate raw data from LEUV and IPN1 was supported by the grant no. 18-05-00343 from the Russian Foundation for Basic Research.

**Acknowledgments:** GPS data from UCOE and SPIG stations was provided by the Servicio de Geodesia Satelital (SGS) TLALOCNet network [21] and the Servicio Sismológico Nacional at the Instituto de Geofísica-Universidad Nacional Autónoma de México (UNAM) [22]. The authors express their gratitude to the SGS and SSN personnel, in particular to Luis Salazar-Tlaczani at SGS. The authors thank the support team of Septentrio (www.septentrio.com), in particular Gustavo Lopez Andrade and Yasmine Hunter. The authors also thank Instituto Politecnico Nacional (IPN), in particular Miguel Sanchez Meraz from the Escuela Superior de Ingenieria Mecanica y Electrica (ESIME) of IPN, for the provided raw data of IPN1 station. GPS data from ISTP station was provided by SibNet network [23] of ISTP SB RAS Angara Common Use Center (http://ckp-rf.ru/ckp/3056). The authors would like to thank the editor and the anonymous reviewers for their contributing comments.

**Conflicts of Interest:** The authors declare that they have no conflict of interest.

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
