# Peer review of "Comparison of TEC Calculations Based on Trimble, Javad, Leica, and Septentrio GNSS Receiver Data"

_remotesensing, doi:10.3390/rs12193268_

Round 1
Reviewer 1 Report
The authors present a comparison study on the accuracy of slant TEC calculation to reveal how TEC noise component depends on receiver types/models. Four types of receivers are considered, including Javad, Septentrio, Trimble and Leica GNSS receivers. In this work, the TEC for each type of receivers is calculated based on L1 and L2 carrier phase data.
Both an analytical modelling and real data of the TEC noise component were considered for the comparison. In addition, correlation analysis between L1 and L2 carrier phase noises was proposed to help understand L2 signal tracking technique implemented in the receivers.
This research topic is practical of importance considering there are a diversity of GNSS receivers/measurements applied for ionospheric studies. Differences among receivers in TEC and TEC-based indices may cause incorrect interpretation of ionospheric observations. A good understanding of the receiver performance on TEC calculation is essential to ionospheric studies.
Overall, the paper is well written. Below listed are major concerns:
- Experimental data were only for quite geomagnetic time (Table 1). It is unclear how the TEC noise changes with respect to each type of receivers under disturbed conditions. Additional experiment needs to be considered to demonstrate it.
- The analytical model of the TEC noise component is presented (EQ. 5) but without details describing how the model is developed. As the carrier phase noise is calculated in case of the time resolution > 10 Hz, does this model work for all time resolutions (i.e., 1 Hz, 5 Hz, …)? Please clarify.
- Correlation analysis/coefficient between L1 and L2 carrier phase noises was proposed to learn if L1-aiding technique is used to track L2 signal (Fig. 2). However, it does not very help explain the dependent of the TEC noise on receiver types/models (Fig 3 and Fig 4). So what is the main contributor? Any discussion should be elaborated in the paper.
Other comments:
Table 1: Please specify the receiver type/model. There are a number of receiver models for one manufacturer.
Figure 1 and Figure 3:
Their captions are informative. Are L2P and L2C cases both shown in the plot for Trimble receivers? Please clarify.
Line 186: “0.0075 TECU” should be “0.00075 TECU”?
Lines 192-194:
The data for Leica receiver was not on the same day as for other receivers. Is this a factor causing the difference?
Author Response
We are very appreciated the Editor and the Reviewers for their valuable comments. The manuscript was revised according to your suggestions. Attached please see our answers to the Reviewer’s 1 comments.

Reviewer 2 Report
Nowadays the TEC calculations are obtained from networks of GNSS receivers since are more available than ionosondes. This paper compares TEC values calculated by different receivers, with the aim of proving the influence of the receiver's models and TEC accuracy. It is easy to understand that this "letter" is very important for researchers. I have only a few doubts, which are concerning data and materials, and I would like the authors clarifying:
1) Despite being a topic that fits in the Remote Sensing journal, I think it would have more reach, impact, and interest for other researchers, if this paper was presented to other journals (more related to ionosphere studies and their impact on the quality of the GNSS signal). So, I ask the authors for the reason/reasons for not having opted for other journals as "Space Weather", "Journal of Space Weather and Space Climate", "GPS Solutions", among others.
2) I understand the difficulty in acquired data at the same place, day, and time. However, the ionosphere is affected in several ways, being geographical location one of them. The data was collected by different receivers models each one on a different location. How can this affect the results obtained and, consequently, the conclusions?
3) Also, based on the same idea in (2), what will be the conclusions expected, if, at least, two different receivers were installed in the same place?
4) The plots of the figures show differences between the two Trimble receivers. This can be due to the geographic location? Or due to different Trimble receiver models? This is, the Trimble receivers are the same model?
Author Response
We are very appreciated the Editor and the Reviewers for their valuable comments. The manuscript was revised according to your suggestions. Attached please see our answers to the Reviewer’s 2 comments.

Round 2
Reviewer 1 Report
Thanks for the authors' efforts in addressing the concerns. I would like to recommend to accept this paper in present form.
Reviewer 2 Report
The paper addresses a very important issue for researchers. This work compares TEC values ​​calculated by different receivers, in order to prove the influence of the receiver models and the TEC precision. The problem and method used are well exposed and with consistent results.
The authors clarified my doubts and incorporated them into the new version (including suggestions from another reviewer). In my opinion, this article is ready to be published in the special edition “High GNSS rate data for the ionospheric study”.
This manuscript is a resubmission of an earlier submission. The following is a list of the peer review reports and author responses from that submission.